# Swedish High School Students’ Drug and Alcohol Use Habits throughout 2020

**DOI:** 10.3390/ijerph192416928

**Published:** 2022-12-16

**Authors:** Anis Sfendla, Kourosh Bador, Michela Paganelli, Nóra Kerekes

**Affiliations:** 1Higher Institute of Nursing Professions and Health Techniques, Errachidia 52000, Morocco; 2Department of Biology, Faculty of Sciences, Abdelmalek Essaâdi University, Tetouan 93000, Morocco; 3Center for Holistic Psychiatry Research (CHoPy), 431 60 Mölndal, Sweden; 4AGERA KBT, 411 38 Gothenburg, Sweden; 5Department of Health Sciences, University West, 461 86 Trollhättan, Sweden

**Keywords:** adolescents, alcohol, AUDIT, drug, DUDIT, gender, Sweden, cutoffs

## Abstract

This study describes gender-specific patterns in alcohol and drug use among Swedish high school students throughout 2020 and questions the current cutoffs for identifying addiction in this population. From September 2020 to February 2021, 1590 Swedish upper secondary high school students (mean age 17.15 years, age range 15–19 years, 39.6% male, and 60.4% female) completed the anonymous, electronic survey of the Mental and Somatic Health without borders study. The respondents reported their substance use habits during the previous 12 months using the Drug Use Disorders Identification Test (DUDIT) and Alcohol Use Disorders Identification Test (AUDIT). They also answered questions about changes in their alcohol and illegal drug use habits after the COVID-19 outbreak. No gender differences were detected in the prevalence and degree of alcohol use. Compared to female adolescents, significantly more male adolescents used drugs (and to a significantly higher degree, although with a small effect size). Substance use problems peaked in females at age 17 and in males at age 18. The COVID-19 outbreak affected alcohol consumption and illegal drug use in male and female adolescents similarly. For both genders, of those who used illegal drugs, over 40% reported increased use after the outbreak. Our results reinforce previous suggestions of the narrowing of gender differences in Swedish adolescents’ risk behaviors and challenge the previously validated gender-specific cutoffs for the AUDIT and DUDIT. An improved understanding of the impacts of gender diversity and evolving gender roles and norms on behaviors and mental health is warranted.

## 1. Introduction

### 1.1. Adolescence as a Critical Period for Development

Adolescence represents the major life transition from childhood to adulthood [1], during which biological process-based psychological development elicits pubertal maturation [2]. The neurobiological and hormonal changes associated with puberty entail profound cognitive and behavioral development: an adolescent develops a new perception of body image and a new sense of self and adapts their social and emotional skills during a gradual shift toward adulthood [3]. One hallmark of adolescent behavior is attraction to novelty [4]. This is hypothesized to be an ontogenetic adaptation that fosters exploratory behaviors and facilitates the maturation of new skills, which potentially enable an adolescent to reach independence from caregivers [5]. However, in addition to exploration and adaptation to the challenging borderland between childhood and adulthood [6], novelty seeking may result in an increased propensity to engage in risk-taking behaviors [4]. Adolescents tend to be more prone to risk-taking behaviors—often expressed through norm-breaking behaviors, including delinquency and the use of alcohol and illicit drugs [6]—than individuals of other age groups [7].

### 1.2. Substance Use in Adolescents

Initial experimentation with alcohol, tobacco, and illicit drugs typically debuts during adolescence [8]. This behavior is associated with adverse short-term consequences (e.g., intoxication, risky sexual and driving behavior, physical injuries) along with undesired long-term effects on adolescents’ health (e.g., the development of addiction and increased morbidity and mortality) [9] and future life options (e.g., social rejection or isolation, legal troubles, financial losses, and hindered educational and occupational goals) [4].

According to the latest report from 2020 [10], in Sweden, alcohol and illicit drug use among 15- and 16-year-old school students was considerably lower than the European average. Sixty-five percent of 15- and 16-year-old Swedish students reported using alcohol at some point in their lives (as opposed to 80% of European students). Furthermore, 26% reported using it in the prior 30 days and 9% reported at least one episode of intoxication following heavy drinking in the prior 30 days (the average European percentages for the 30-day use and intoxication rates were 48% and 13%, respectively). Rates of drug use vary between European countries; an average of 8% of Swedish students have tried an illicit drug at least once during their lifetime (without any significant gender differences), compared to a European average of 18% (ranging from 15% of girls to 21% of boys) [11].

In 2015, however, Swedish students did not significantly differ from the European average with respect to lifetime use of psychoactive pharmaceuticals without prescriptions: 3% of Swedish students reported using painkillers and 7% reported using tranquilizers and/or sedatives to become affected, compared to the European averages of 4% and 6%, respectively [11]. In the same year, no notable differences were observed in the use of inhalants or new psychoactive substances, whose rates displayed the same averages in Sweden and Europe (7% for the use of inhalants and 4% for new psychoactive substances, without any emerging gender differences) [11].

In Sweden, compulsory care is regulated under the Care of Persons with Substance Use Disorders in Certain Cases Act (SFS 1988:870) and the Compulsory Psychiatric Care Act (SFS 1991:1128). Though coercive treatment is in general used rarely, it is employed rather more often in the case of alcohol and drug dependence. Additionally, therapeutic communities have played a role traditionally and are qualified as the most popular treatment modality in Sweden. In a recent years, outpatient treatment became more adopted and is based on the 12-Steps model [12,13,14].

Gender differences in the prevalence, risks, and consequences of substance use are well established [15]. Women are reported to have higher risk, higher consumption amounts, accelerated onset of substance use disorder, and more serious consequences [16]. However, recent studies from Sweden have indicated that there is only a small gender difference in adults who seek treatment [17]. Furthermore, data from both the European School Survey Project on Alcohol and Other Drugs and the Swedish Council for Information on Alcohol and Other Drugs (Centralförbundet för alkohol- och narkotikaupplysning) indicate that socialization processes are narrowing the traditional gender gap in substance use among Swedish adolescents, especially with respect to alcohol consumption [11,18,19]. In recent years, among the Swedish female adolescent population, there has been a substantial impact on mental health and the perception of hostile environments. In a society (Sweden) that has long been influenced by feministic movements, gender differences (even in disruptive behavior patterns) have decreased, and Swedish female students have shown increased levels of coping strategies that have previously been associated with the male gender (e.g., hostility) [20].

### 1.3. The Impact of COVID-19 Restrictions on Adolescents’ Substance Use

The COVID-19 pandemic and related preventive measures to stem its spread introduced profound changes in day-to-day life, thus affecting the mental health of the global population [21].

It has been suggested that adolescents who use substances are particularly vulnerable to the effects of the pandemic [22]. Adolescents already present a vulnerability since they are living in a critical development period [23], and the changes in day-to-day life imposed by the COVID-19 pandemic may increase this vulnerability, potentially leading to the increased risk of substance use. Research on this potential correlation is still limited and focused on adults [22]. However, the studies investigating adolescents have indicated that the COVID-19 pandemic is a potential trigger of risk factors that may increase the vulnerability of the adolescent population [24,25], especially those adolescents who are already substance users [22].

Nevertheless, increased vulnerability is not always accompanied by the increased use of substances. Owing to restrictions and limitations, adolescents might have reduced access to alcohol and fewer possibilities to consume it (e.g., because of strict restrictions for meeting in groups and having celebrations and parties). This may be why there was a larger proportion of adolescents who decreased their alcohol use during the pandemic than of those who increased it [22], even though the vulnerability that predisposed them to this behavior may have increased [26].

In Sweden, no strict COVID-19-related restrictions went into effect. The approach taken as a response to the pandemic has been called “soft-touch” or “light-touch”. Working from home was recommended, not imposed, together with the avoidance of “unnecessary travels” using public transportation. High school and university lessons were delivered online. The only legal restrictions were limited to venues serving food and drinks (i.e., only seated guests could be served), and public places could not hold events with more than 50 people [27].

Considering the above context, the present study aimed to provide an update on male and female Swedish upper secondary high school students’ self-reported alcohol and drug use habits during 2020 and to investigate the changes in these habits associated with the COVID-19 outbreak.

We hypothesized that the gender-based differences in substance use in this population would be further decreased and that the COVID-19 outbreak would have similarly impacted both genders’ substance use.

## 2. Materials and Methods

### 2.1. Study Design and Procedure 

The data analyzed in the present study were derived from the cross-sectional, multinational study “Mental and Somatic Health without borders” (MeSHe; www.meshe.se (accessed on 5 December 2022)). The MeSHe survey consists of several validated questionnaires through which young people rate their own mental and physical health and risk behaviors. A detailed description of the MeSHe study can be read at www.meshe.se (accessed on 5 December 2022) and [20]. For the present study, the relevant questionnaires from the electronic version of the MeSHe survey were the Drug Use Disorders Identification Test (DUDIT) [28] and the Alcohol Use Disorders Identification Test (AUDIT) [29].

High school students in Sweden were contacted via their schools and through social media. The survey provided the students with information about available national (Swedish) resources and support organizations.

From September 2020 to February 2021, we collected 1662 responses. Of these, 292 (mean age 17.31, SD = 0.78, 62% female, 37% male, and 1% other gender identity or missing response) were received during the first months through direct contact with high school administrations. These were mostly from the cities of Uddevalla and Stockholm. Because of the low response rate using direct contact with high schools, during the 2020 Christmas holidays, the survey was announced on social media, specifically targeting 15- to 19-year-old students through the use of the “Evasys” course evaluation and survey system (https://evasys.de/en/ (accessed on 18 February 2021)).

There were 10,693 (59% males and 41% females) active clicks on the link, which generated 1370 responses (a 12.81% response rate). Responses were received from all 21 counties in Sweden. Those outside the age range of 15–19 years (*n* = 55) were excluded from the dataset, resulting in a working data file of 1607 responses (mean age 17.12, SD = 0.96, 59% female, 40% male, and 1% other gender identity or missing response). The number of gender non-binary students (*n* = 15) was too low to form a separate group for statistical analysis; therefore, respondents identifying themselves as non-binary were also excluded from the analysis. Outliers with repetitive maximal responses on all items (*n* = 2) were also excluded for measurement error purposes. The final study population included 1590 Swedish high school students (39.6% male and 60.4% female) with a mean age of 17.15 (SD = 0.88) years.

### 2.2. Instruments 

#### 2.2.1. Drug Use Disorder Identification Test (DUDIT)

The DUDIT contains 11 questions that use self-reports to capture the frequency of drug use during the prior year. Items 1–9 are scored on a 5-point Likert scale, while items 10 and 11 are scored on a 3-point Likert scale. The total DUDIT score, calculated by the sum of all item responses, ranges between 0 to 44. Drug-related problems are identified by cutoffs of ≥6 for men and ≥2 for women [28]. The DUDIT has been previously used in adolescent populations [30,31]. The Cronbach’s alpha for the DUDIT in the present study was 0.94. A recent psychometric evaluation of the DUDIT suggested a two-factor model for screening in adolescents. Factor 1 (items 1–4) refers to consumption, with scores between 0 and 16. Factor 2 (items 5–11) refers to drug-related problems with a possible score ranging between 0 and 28 [32].

#### 2.2.2. Alcohol Use Disorder Identification Test (AUDIT)

In 1993, the World Health Organization developed the AUDIT [29], intended to identify early hazardous and harmful drinking. The instrument contains 10 items, each scored on a 5-point Likert scale except questions 9 and 10, which can score 0, 2, or 4 points, with a total score ranging from 0 to 40. Items 1–3 focus on consumption and items 4–10 are about problems related to alcohol habits [33]. Cutoffs of ≥8 for men and ≥6 for women identify the harmful or hazardous use of alcohol [29]. The AUDIT has previously been used to assess alcohol problems among the general population [34] and adolescents [31,35]. The Cronbach’s alpha for the AUDIT in the present study was 0.78.

#### 2.2.3. COVID-19-Related questions

During the 2020 data collection for the MeSHe project, the e-MeSHe survey included COVID-19-related questions [25]. In the present study, we used two of these questions to compare the responses on the AUDIT and DUDIT addressing the proportion of adolescents who had never used substances. The purpose of this was to pinpoint the generalizability of the results based on the adolescents’ reports of decreased or increased alcohol and illegal drug use during the pandemic. These two questions from the COVID-19-related questionnaire both asked: “Look back at and compare your behaviour before the COVID-19 outbreak and your present behaviour. Has it changed? If so, how? To what extent the change corresponds best to your situation?” The first iteration of the question involved “Consuming alcohol”, and the second iteration involved “Illicit drug use: including prescription drugs used for different reasons other than prescribed.”

### 2.3. Statistical Analysis

Descriptive statistics (mean (M), standard deviation (SD), and frequency (%)) were used at both the scale and item levels. The distribution of the data was tested using the Kolmogorov–Smirnov test and showed a significant difference from the normal distribution (*p* < 0.001). This non-normal distribution was supported also by the values of skewness and kurtosis (for DUDIT total score the skewness was 5 and kurtosis was 34, while for AUDIT total score these were 2 and 4, respectively) [36]. Hence, non-parametric tests were used in the analyses. The Mann–Whitney U test was used to analyze differences between male and female students’ AUDIT and DUDIT factors, items, and total scores. Furthermore, η^2^ was used to report effect size. The following equation was used: η^2^ = Z^2^/(N − 1)
where η^2^ > 0.01 indicates a small effect, η^2^ > 0.06 indicates a medium effect, and η^2^ > 0.14 indicates a large effect. The significance level was set at *p* < 0.05. All analyses were performed using SPSS version 28 (IBM).

### 2.4. Ethical Considerations

The regional ethics review board in Gothenburg approved the project (registration number 689–17). This was updated in 2020 by the National Ethical Board after the COVID-19-related questions were included in the survey and the survey was changed to an electronic version (registration number Dnr: 2020-03351). The completion of the electronic survey was voluntary and anonymous; respondents provided typed informed consent before completing the survey.

## 3. Results

### 3.1. Swedish High School Students’ Alcohol Use Habits during 2020

The total AUDIT scores from 1483 students were analyzed; 107 students (6.73%) were excluded from the frequency analysis due to missing answers on one or more items. There were 556 students (37.5%) who scored zero total points on the AUDIT, indicating never using alcohol. The mean total AUDIT score was 2.89 (SD = 3.61), with scores ranging from 0 to 24. The distribution of male and female Swedish students’ total AUDIT scores (those >0) is shown in Figure 1.

There was no significant difference detected between the proportions of male and female students with respect to engagement in alcohol use. Of the respondents, 39% of male and 36.5% of female students reported no alcohol use (*n* = 227 and *n* = 329, respectively; *p* = 0.45). Male students reported more serious engagement in alcohol use (total AUDIT scores of M = 3.30 and SD = 4.02 in males and M = 2.98 and SD = 3.46 in females), but this difference was not significant (*p* = 0.75; Table 1).

Alcohol use habit comparisons at the factor level revealed no significant difference between male and female Swedish high school students (alcohol consumption factor *p* = 0.139 and alcohol problem factor *p* = 0.975).

When comparing alcohol use habits at the item level, significant differences were found for three items: item 2 (“How many drinks containing alcohol do you have on a typical day when you are drinking”), *p* = 0.004, η^2^ = 0.005; item 3 (“How often do you have six or more drinks on one occasion?”), *p* = 0.012, η^2^ = 0.004; and item 6 (“How often during the last year have you needed a first drink in the morning to get yourself going after a heavy drinking session?”), *p* = 0.024, η^2^ = 0.003. For each of these items, male students reported significantly higher scores than females, albeit with negligible effect size (Table 1).

Using the previously suggested gender-specific cutoffs (≥8 for men and ≥6 for women [37]), 13% of male students and 15% of female students were identified as displaying harmful or hazardous use of alcohol (Table 2). Table 2 also indicates the prevalence of adolescents with harmful or hazardous use of alcohol if the same (non-gender-specific) cutoff was to be applied.

Applying a non-gender-specific cutoff of ≥6 resulted in the earlier detection of harmful or hazardous alcohol use in male students. A non-gender-specific (any) cutoff represented more realistic harmful or hazardous alcohol use among Swedish high school students, as no significant differences in total scores were detected.

### 3.2. Swedish High School Students’ Drug Use Habits in 2020

Of the participants, 15.6% (*n* = 248) did not answer at least one question in the DUDIT inventory; therefore, their total DUDIT scores could not be calculated and were classified as missing values in our analyses. A final sample of *n* = 1342 was included for the following analyses.

The mean value total DUDIT score was 0.81 (SD = 2.84, range 0–29). The majority of students (84.9%) reported a total score of zero points, indicating that they had never used any type of illegal drugs. Just over 15.1% (203 students) reported varying degrees of drug use habits (Figure 2).

According to the total DUDIT scores, there was a small but significant effect size (*p* < 0.001, η^2^ = 0.027) difference between male students’ (total DUDIT score M = 1.38, SD = 3.81) and female students’ (total DUDIT score M = 0.46, SD = 1.92) drug use habits. The proportions of male and female students engaging in drug use also differed significantly (*p* < 0.001), with a higher proportion of male students (*n* = 116; 22.3%) than of female students (*n* = 87; 10.6%) indicating this behavior.

The first four items that focus on drug consumption (factor 1) revealed a significant difference between male (M = 0.72, SD = 1.73, *n* = 543) and female students (M = 0.23, SD = 0.86, *n* = 848) with a small effect size (*p* < 0.001, η^2^ = 0.030). The drug-related problems factor (factor 2) also revealed a significant difference between males (M = 0.65, SD = 2.45, *n* = 529) and females (M = 0.24, SD = 1.25, *n* = 834), but hardly reached a small effect size (*p* < 0.001, η^2^ = 0.010).

Male participants scored significantly higher (indicating a higher frequency of that behavior) than their female classmates for each item in the DUDIT except item 8 (“How often over the past year have you needed to take a drug the morning after heavy drug use the day before?”), for which the difference was very close to significant (*p* = 0.05; Table 3). The effect size of the difference was small for the first four items and negligible for the rest of the items.

Even with these significant differences between genders, when the previously validated cutoffs were applied (DUDIT ≥ 2 for females and ≥6 for males), we could identify 9% of female students and 8% of male students with drug-related problems in the sample. Table 4 shows the frequency of drug-related problems if a gender-neutral cutoff was to be applied. If the lower cutoff (DUDIT ≥ 2) was to be used for both genders, then 20% of the male students and 9% of the female students would be identified as having drug-related problems. If the higher cutoff (DUDIT ≥ 6) was to be used for both genders, then 6% of the male students and 2% of the female students would be recognized as having problematic drug use.

### 3.3. COVID-19-Related Risk Behaviors

In another part of the MeSHe survey, adolescents reported how the COVID-19 pandemic affected their alcohol and illicit drug use habits. Similar proportions of male and female adolescents (39.5%) reported that they did not consume alcohol before or after the COVID-19 outbreak (Table 5). This proportion is noticeably similar to that of adolescents who scored zero total points on the AUDIT (37.5%), indicating that they had never consumed alcohol during the prior 12 months. Among those who did report alcohol consumption, similar proportions of male and female respondents (*p* = 0.40) indicated that their alcohol use increased, decreased, or did not change after the outbreak (Table 5).

The proportion of adolescents reporting no illegal drug use at all on the COVID-19-related questionnaire (88.7%) was comparable to the proportion detected by the DUDIT (84.9%). Furthermore, the gender pattern was similar to that found with the DUDIT; for example, significantly more female students than male students reported no illegal drug use (*p* < 0.001). Of those who indicated illegal drug use, noteworthy proportions (44.3% of males and 42.9% of females) reported increased use since the outbreak (Table 5). 

## 4. Discussion

This study identified self-reported substance use habits among Swedish high school students during 2020 and explored gender-specific patterns. The results should be interpreted with an understanding of the environmental changes surrounding upper secondary high school students during 2020. Worldwide, COVID-19 restrictions were actualized, and behaviors (including risk behaviors) naturally changed as students adapted to the new environment. Many researchers predicted and reported negative long-term consequences among children and adolescents—particularly excessive recreational screen time, poor diet, physical inactivity, and poor sleep, but also increases in substance use, the prevalence of psychiatric disorders, and suicidal thoughts [38,39,40,41,42]. However, a few reported that restrictions actually resulted in decreases in alcohol consumption and norm-breaking behaviors and that adolescents showed resilience [25,43,44]. Hafstad and Augusti [45], in their comment in The Lancet Psychiatry in 2021, drew attention to the fact that changes in adolescents’ mental health and risk behaviors during the pandemic cannot and should not be simply interpreted as consequences of the COVID-19 pandemic and its restrictions—increased psychological distress levels could be measured in adolescents even before the pandemic. Nguyen et al. [46] also showed that the levels of psychological distress and the impacts of the pandemic on adolescents’ well-being varied greatly across countries and cultures. Time and environment, both separately and together, impact well-being and behaviors; therefore, the results of the present study should be interpreted both within a bigger perspective (possible generational changes in risk behaviors) and in a more restricted matter (only applicable to Swedish adolescents).

Understanding patterns of substance use between genders remains a great challenge. It is well established that in addition to biological aspects, strong socio-cultural influences (such as gender equality and income and changes in working patterns) impact and modify risk behaviors [47]. Included in these socio-cultural factors are changes in gender terminology and gender roles and the diversification of gender identities during recent decades [35]. In a culture and healthcare environment that is still cis-normative, the well-being and psychosocial maturation of adolescents who are not cis-gender are more strongly and negatively affected than those of their cis-gender peers [48]. Increased knowledge of and respect for gender diversity and the acknowledgment of evolving, declining gender differences (which are more pronounced in some cultures than in others) are necessary in all aspects of society, including revalidation of screening instruments. 

### 4.1. Alcohol Use among SWEDISH Adolescents during 2020

The proportion of Swedish adolescents who had consumed alcohol during the prior 12 months decreased from 89% in 2014 to 69% in 2019 [49]. The present study shows a further decrease (to approximately 60%) in 2020. Such a decrease should be interpreted with caution, as decreased use does not necessarily mean non-problematic levels of alcohol use. Additionally, this decreased use might be related to decreased in-person social interactions, since social distancing was imposed during lockdown in 2020. Another explanation could be related to peers’ perceptions: a change in behavior is usually sought if people believe that others are changing their behavior too [50]. It was previously suggested that the reasons behind adolescents’ drinking habits were very similar to those indicated by adults (e.g., sociability, relaxation, companionship, excitement, peer pressure, and a tradition of drinking in their community) [51]. A society that includes alcohol as a part of cultural socialization will always have youth who imitate the drinking behaviors of adults as a natural part of the transformation to adulthood.

The present sample revealed similar proportions of male and female students who engaged in alcohol use. While male students reported somewhat more severe alcohol use habits, this difference did not reach the level of significance. These results indicate that adolescents demonstrate a similar pattern to that previously seen in adults—in 2009, it was shown that there was a narrowing of the traditional gender gap with respect to alcohol consumption in the adult populations of several countries. For example, the ratios of male-to-female drinkers aged 18–34 years were 1 to 1 in Norway, 1.08 to 1 in Sweden, 0.93 to 1 in New Zealand, and 1.06 to 1 in Japan [52]. In Swedish adolescents, over the past 20 years, a nearly stable pattern was detected: approximately 4% more female students (second-year upper secondary high school) reported alcohol use during the previous months, but male students reported consuming greater amounts of alcohol (p. 17, [43]). AUDIT item-level comparisons in the present study revealed some significant but negligibly sized differences between male and female adolescents’ alcohol use habits. These items were the typical quantity of drinking, frequency of heavy drinking, and morning drinking, which were all more prevalent among males than among females. The notions that male adolescents consume more alcohol and indicate more frequent heavy drinking and greater amounts of alcohol consumption than females are consistent with previous national reports [49]. Morning drinking is related to dependency symptoms and is more often described after heavy alcohol consumption [53], which was previously reported in a higher frequency among males [54]. 

No gender-based differences in alcohol consumption were detected when our results were interpreted within the context of the COVID-19 pandemic. Of the approximately 60% of adolescents who reported alcohol consumption during 2020, among both genders, approximately one-third reported that their consumption did not change, approximately one-third reported that their consumption decreased, and approximately one-third reported that their consumption increased after the outbreak. Similarly, a previous study conducted among Canadian young adults reported no gender-based differences in changes in alcohol use habits; however, a significantly decreased level of alcohol consumption was reported for both genders (mean age 23.48 years old) [55]. However, the picture about gender-specific changes with respect to alcohol use existing or not existing during the pandemic becomes more unclear when we consider the following: within a nationwide sample of 13- to 18-year-old adolescents in Iceland, female adolescents reported a higher level of alcohol intoxication than male adolescents [56]. Furthermore, in a Danish sample of 15- to 20-year-olds, female participants were more likely than male participants to have a perceived decrease in their alcohol use during lockdown [57].

Screening purpose implies, for a stable risk identification, that the selection of risk indicators should be proportional between consumption domain parameters, alcohol use disorder characteristics, and early morbidity in order to generate a solid risk identification performance. As cutoffs of screening instruments are based on T-score analyses from the general population, consumption profiles should be updated more often, most importantly after such dynamic societal changes that the past decennium presented.

### 4.2. Drug Use in a Swedish High School Student Sample

The majority of Swedish students (approximately 85%) reported no use of any type of illegal drugs during the prior 12 months. The proportion of Swedish upper secondary high school students who reported using illegal drugs in the present study (15%) is comparable to (but somewhat higher than) the proportion measured in 2019 (13%) (p. 44, [43]). While a pattern of increasing illegal drug use was already detected during the period of 2006 to 2019 in Swedish adolescents, the circumstances of the pandemic may have exacerbated and sped up this process. This type of change in risk behaviors, such as an increase in illegal drug use during the COVID-19 restrictions, was reported in a multinational sample of adolescents [25]. It was speculated to be dependent on circumstances—such as having fewer possibilities to go out and party or buy alcohol, while still having access to prescription drugs through the internet. However, with respect to illegal drug use among youth during the COVID-19 pandemic, a recently published systematic review concluded that the findings were heterogeneous. Some studies reported increases in drug use (from low-, middle-, and high-income countries), some reported decreases in use, and one study even reported no change during the pandemic [58]. In this systematic review, four studies included mentions of gender, but none of them included it in relation to differences in substance use.

In our study, we found that male adolescents reported significantly more severe drug use than female adolescents, albeit with a small effect size. While the significant differences were shown at the item level, it is important to recognize that only four of these item differences reached a small effect size; the others were negligible. Those four items were the first four questions, focusing on drug consumption [32] and measured frequency of drug use (item 1: “How often do you use drugs other than alcohol?”; item 3: “How many times do you take drugs on a typical day when you use drugs?”), the frequency of heavy drug use (item 4: “How often are you influenced heavily by drugs?”), and the existence of polydrug use (item 2: “Do you use more than one type of drug on the same occasion?”). This indicates that Swedish male adolescents use more types of drugs more often and more seriously than females do, while the consequences of drug use (captured by items 5–11) significantly differed between the genders but without a measurable effect size at item level, and with the lowest verge of a small effect size at a factor level. 

In 2019, in a national report on substance use, it was reported that 19% of male and 13% of female students in their second year of upper secondary school had used drugs. In our study, 22.3% of male respondents and 10.6% of female respondents reported the use of illegal drugs in the previous 12 months, which indicates a much more drastic gender difference than that measured before the COVID-19 pandemic. However, using the previously validated cutoffs for the DUDIT [28], the prevalence rates of drug abuse among the Swedish adolescents in our sample are in line with those from national and European surveys [6], indicating a higher prevalence of females being dependent on substances. Interestingly, in a recently published validation of the DUDIT in a psychiatric adolescent population, only gender-neutral cutoffs were identified for the DUDIT; furthermore, pattern identification was based on its first four items (consumption), for screening use within clinical populations [32]. Our results therefore strengthen the importance of the first four items (consumption factor) of DUDIT in the identification for substance use disorder among adolescents.

We should not speculate about the reasons and explanations for the above, as we did not measure any additional parameters that could be used as descriptors. However, we did measure to what extent male and female adolescents changed their drug use habits after the COVID-19 outbreak. Interestingly and importantly, while the absolute proportion of female adolescents using illegal drugs was significantly smaller than that of male adolescents, the pattern of changes was exactly the same after the outbreak. That is, 40% of those using increased their use, 30% decreased their use, and 30% did not change their habits. Similar trends of enhanced substance use were previously associated with multiplied pressure, fear, imbalance among extraordinary additives of normal existence, social isolation, and stressful occasions, alone and together [59,60].

The fact that an increased proportion of adolescents, independent of their identified genders [48], increased their illegal drug use during 2020 calls for awareness in different fields of society. Are we prepared to detect those at risk and help those in need, regardless of their gender identities? Are gender-specific cutoffs products of a cis-normative culture, where females are more vulnerable by definition? We cannot answer these questions based on the results of the present study, but we hope to awaken awareness about the need for gender-neutral, equal, and person-centered holistic perspectives in care.

## 5. Strengths and Limitations

A major strength of the present study is its considerably large, general study population, from all parts of Sweden, to capture adolescents’ substance use habits during the first year of the COVID-19 pandemic (2020). However, the study population is not a representative sample because of the sampling process that was based on a non-probabilistic sampling process. Among the additional strengths of the current study, the AUDIT and DUDIT have previously shown good psychometric properties; furthermore, they have advanced characteristics over other screening instruments, such as their briefness and abilities to map the past 12 months and screen for quantity, frequency, and societal consequences of substance use. Using anonymous self-reports is a strength, as such respondents may more freely report real information about their using habits without worrying about shame or consequences. However, this is also a limitation, as it means that no objective register or clinical data could control for the truthfulness of the information. Importantly, retrospective perceptions of changes are likely not identical to changes that might have been observed in a longitudinal prospective design. Another limitation is the cross-sectional design of the study, which limits causality and developmental analyses. This could be more plausible in the present study if we had defined current specific cutoff scores to shed light on the actual gap between previous and current alcohol and drug use patterns. However, a lack of clinical and socioeconomic information about the respondents precluded this possibility.

## 6. Conclusions

No differences were detected between genders with respect to alcohol use among Swedish adolescents between the ages of 15 and 19 during 2020. The COVID-19 outbreak was not associated with a change (increase or decrease) in alcohol consumption in male or female adolescents. 

Compared to female respondents, approximately 15% more male respondents reported illegal drug use during 2020. Gender differences were small for items describing frequencies and negligible for items describing consequences. After the COVID-19 outbreak, approximately 40% (of both genders) who previously used drugs increased their use.

These results challenge the gender-specific cutoffs that are used for screening instruments among Swedish adolescents and argue for improved knowledge as well as gender-neutral screening and treatment for risky behaviors among this population. 

## Figures and Tables

**Figure 1 ijerph-19-16928-f001:**
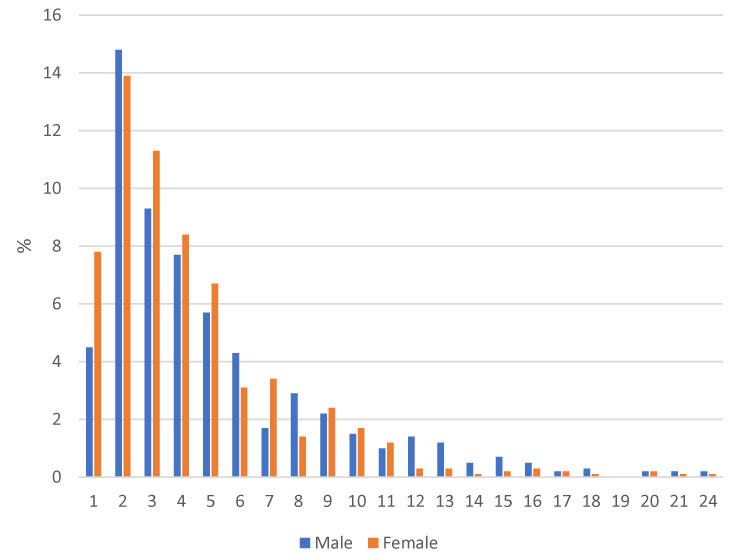
Frequency of total AUDIT scores in the Swedish high school student sample by gender (*n* = 927), excluding the 37.5% of the sample that indicated no alcohol use at all (scored as 0).

**Figure 2 ijerph-19-16928-f002:**
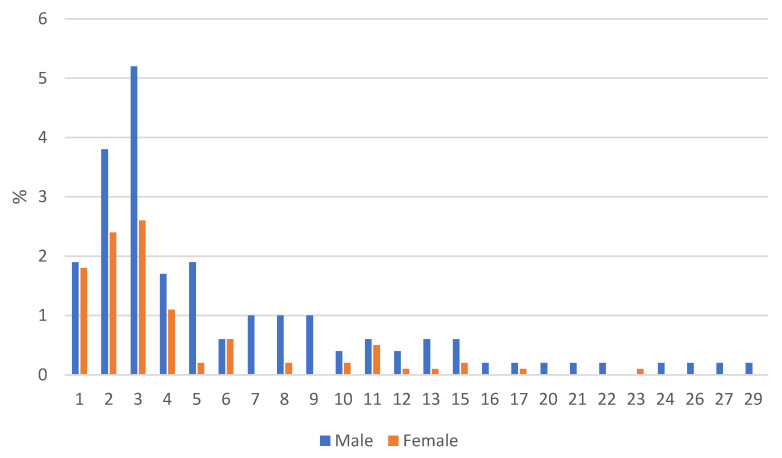
Frequency of total DUDIT scores in Swedish high school student sample by gender (*n* = 203), excluding the 84.9% of the sample that indicated no drug use at all (scored as 0).

**Table 1 ijerph-19-16928-t001:** Comparison of scores for each AUDIT item and total AUDIT scores between male and female Swedish high school students.

Item	Female Students			Male Students			*p*
	*n*	Min–Max	M (SD)	*n*	Min–Max	M (SD)	
1. How often do you have a drink containing alcohol?	930	0–3	0.89 (0.80)	606	0–4	0.92 (0.88)	0.79
2. How many drinks containing alcohol do you have on a typical day when you are drinking?	919	0–1	0.01 (0.11)	600	0–1	0.03 (0.18)	0.004
3. How often do you have six or more drinks on one occasion?	930	0–3	0.64 (0.75)	604	0–4	0.75 (0.82)	0.012
4. How often during the last year have you found that you were not able to stop drinking once you had started?	927	0–4	0.14 (0.46)	603	0–3	0.12 (0.44)	0.32
5. How often during the last year have you failed to do what was normally expected of you because of drinking?	925	0–4	0.18 (0.47)	601	0–4	0.19 (0.48)	0.86
6. How often during the last year have you needed a first drink in the morning to get yourself going after a heavy drinking session?	929	0–3	0.03 (0.22)	601	0–3	0.06 (0.31)	0.024
7. How often during the last year have you had a feeling of guilt or remorse after drinking?	929	0–3	0.27 (0.54)	602	0–3	0.26 (0.55)	0.64
8. How often during the last year have you been unable to remember what happened the night before because of your drinking?	926	0–4	0.31 (0.54)	601	0–3	0.34 (0.59)	0.49
9. Have you or someone else been injured because of your drinking?	936	0–4	0.23 (0.85)	601	0–4	0.27 (0.92)	0.38
10. Has a relative, friend, doctor, or other health care worker been concerned about your drinking or suggested you cut down?	941	0–4	0.13 (0.68)	603	0–4	0.20 (0.84)	0.06
AUDIT Total	901	0–24	2.77 (3.39)	582	0–24	3.07 (3.94)	0.75

*n* = Number of participants; Min = minimum; Max = maximum; M = mean; SD = standard deviation; *p* = significance; AUDIT = Alcohol Use Disorder Identification Test.

**Table 2 ijerph-19-16928-t002:** Distribution of alcohol-related problems based on gender when applying cutoffs of 6 and 8 points on AUDIT total scale (*n* = 582 males; *n* = 901 females).

	Cutoff	*n*	Prevalence
	(≥)		(%)
**Males**	6	111	19%
	8 *	76	13%
**Female**	6 *	139	15%
	8	80	9%

*n* = Number of students with alcohol-related problems/dependence.* Previously validated cutoffs of ≥8 for men and ≥6 for women [37].

**Table 3 ijerph-19-16928-t003:** Comparison of scores on each DUDIT item and total DUDIT scores between male and female high school students.

Item	Female Students			Male Students			*p*	η^2^
	*n*	Min–Max	M (SD)	*n*	Min–Max	M (SD)		
1. How often do you use drugs other than alcohol?	943	0–4	0.08 (0.34)	607	0–4	0.24 (0.66)	<0.001	0.020
2. Do you use more than one type of drug on the same occasion?	869	0–2	0.02 (0.13)	563	0–3	0.07 (0.29)	<0.001	0.013
3. How many times do you take drugs on a typical day when you use drugs?	869	0–3	0.07 (0.28)	553	0–4	0.21(0.52)	<0.001	0.033
4. How often are you influenced heavily by drugs?	861	0–2	0.05 (0.24)	554	0–3	0.17 (0.58)	<0.001	0.026
5. Over the past year, have you felt that your longing for drugs was so strong that you could not resist it?	864	0–4	0.04 (0.24)	553	0–4	0.10 (0.49)	0.014	0.004
6. Has it happened, over the past year that you have not been able to stop taking drugs once you started?	858	0–2	0.01 (0.10)	550	0–4	0.06 (0.40)	0.001	0.009
7. How often over the past year have you taken drugs and then neglected to do something you should have done?	858	0–4	0.03 (0.22)	544	0–4	0.07 (0.35)	0.003	0.006
8. How often over the past year have you needed to take a drug the morning after heavy drug use the day before?	858	0–2	0.01 (0.10)	548	0–3	0.02 (0.19)	0.05	0.003
9. How often over the past year have you had guilt feelings or a bad conscience because you used drugs?	850	0–4	0.05 (0.31)	540	0–4	0.13 (0.53)	<0.001	0.009
10. Have you or anyone else been hurt (mentally or physically) because you used drugs?	861	0–4	0.04 (0.39)	553	0–4	0.09 (0.56)	0.030	0.003
11. Has a relative or a friend, a doctor or a nurse, or anyone else, been worried about your drug use or said to you that you should stop using drugs?	860	0–4	0.06 (0.44)	552	0–4	0.17 (0.73)	<0.001	0.009
DUDIT Total	822	0–23	0.46 (1.92)	520	0–29	1.38 (3.81)	<0.001	0.027

*n* = Number of participants; Min = minimum; Max = maximum; M = mean; SD = standard deviation; *p* = significance; DUDIT = Drug Use Disorder Identification Test.

**Table 4 ijerph-19-16928-t004:** Distribution of drug-related problems based on gender when applying the previous suggested cutoffs of 2 and 6 points on the DUDIT total scale.

	Cutoff	*n*	Prevalence
	(≥)		(%)
Male students	2	106	20%
	6 *	40	8%
Female students	2 *	72	9%
	6	20	2%

*n* = Number of students with drug-related problems/dependence. * Previously validated cutoffs of ≥6 for men and cutoff ≥2 for women [28].

**Table 5 ijerph-19-16928-t005:** Proportions of male and female adolescents reporting whether the COVID-19 outbreak changed their alcohol and illicit drug use habits.

		I Did Not before and Haven’t Started after	Decreased Since the Outbreak	About the Same as before the Outbreak	Increased Since the Outbreak
Consuming Alcohol	Male (*n* = 599)	40.2%	23.4%	18.4%	18.0%
	Female (*n* = 929)	39.1%	23.5%	20.2%	17.1%
Illicit drug use	Male (*n* = 599)	84.1%	4.1%	4.7%	7.0%
	Female (*n* = 937)	91.6%	2.1%	2.7%	3.6%

## Data Availability

Original data tables used for statistical analyses are available by contacting the principal investigator (N.K.).

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
