# Peer review of "Swedish High School Students’ Drug and Alcohol Use Habits throughout 2020"

_ijerph, 2022, doi:10.3390/ijerph192416928_

Round 1
Reviewer 1 Report
Good article, relevant and novel, striking results, and highlighting the update of the bibliographical references. It is true, however, that the analysis of data by factors and not necessarily by items might have provided more adequate results, which would facilitate the interpretation of the results and allow for more solid conclusions. In an evaluation instrument, the individual items do not themselves reflect decisive information, unless the study was of a psychometric nature. However, a study, whose evaluation tools have been previously validated , could justify an analysis of data based on its component factors. If the authors persist in their interest in the analysis of the items, they should justify it in the procedure.
Author Response
First of all, thank you for your supporting and encouraging words and constructive critics.
We have done our best to address each suggestions/comments. In the enclosed document we addressed each of your questions separately, the responses are highlighted in grey throughout the document following the initial suggestions in italics. Thank you for your time and professional impact on our study.
Please, note that the references to pages and lines in the reviews are according to this re-submission, and may possibly not coincide with the pages and lines of the earlier submission.

Reviewer 2 Report
This is a well-written, but purely descriptive paper on substance use among a sample of Swedish high school students in the middle of the corona pandemic. The findings are interesting given Sweden’s different approach to the pandemic, but in absence of a reference group or category it is difficult to interpret the findings from this study.
Throughout the paper, it appears that Sweden does not only have a special position in relation to the corona pandemic, but also in terms of substance use – with much lower prevalence rates than other EU countries. More background information is needed, e.g. regarding the Swedish drug policy and the possibility of compulsory treatment which may refrain many people (youngsters) from using substances.
The cross-sectional study design does not seem the most appropriate design to study changes in substance use habits. Also, recruitment of the study sample is unclear as part of the sample was recruited through high schools and most respondents were recruited through social media announcements. While the social media strategy contributed to an increase in the number of study respondents, it is unclear how this sample relates to the previously recruited sample in terms of age, type of school (vocational vs. general education), SES, .. Overall, background information on the study sample is missing and only 2 or 3 measures are reported (AUDIT, DUDIT + Covid-questions). These instruments are elaborated in great detail at item level, comparing boys and girls. While this may reveal interesting insights, the paper does not look beyond male-female comparisons although other (contextual) variables may play an important role (e.g. urban vs rural, type of school/education, SES).
The application of an alternative cut-off score for the AUDIT and DUDIT may probably result in earlier detection of harmful use, but is probably likely to result in much more false positives or youngsters falsely labelled as ‘at risk users’. Based on this study, it is not possible to make firm conclusions about an alternative cut off point for the AUDIT and DUDIT as an international comparative design is recommended for this type of studies.
The discussion brings in several new elements (e.g. hostility) which have not been reported in the results section and are therefore purely speculative.
Minor remarks:
- Ln 50: the authors refer to negative and adverse consequences of substance use, while adolescents do not start taking drugs for these consequences but for its (short-term) pleasant and positive effects
- A language check is recommended for some odd sentences/phrases
-
Author Response

(The authors gave the same response as above.)

Reviewer 3 Report
Dear Authors,
Thank you for your manuscript. The paper is very well-written and well-structured. The Introduction section presents information on risky behaviours prevalence in adolescents and explains possible reasons for the decreased gap and similar prevalence of alcohol and substance use in gender groups. The Introduction is divided into three subsections. The last subsection explains possible changes in alcohol and substance use during the COVID-19 pandemic. I had some questions raised, but further, I found all the answers provided.
The Methods section contains a description of the study organization and study instruments. It would be helpful to indicate a survey platform used.
Methods used for statistical analyses are clearly described. Shapiro-Wilk test for distribution normality testing makes sense for a limited sample size (n<50 or <100 according to different references). For a large sample, even a small difference from the normal distribution is demonstrated as statistically significant. Thus, the authors should rely on Skewness, Kurtosis and outliers testing to conclude the distribution normality.
The results are clearly presented and very well-discussed. The only comment I have is to provide p-values for the comparison of the proportions meeting the cut-offs for problematic alcohol and substance use across gender groups (if it is possible).
Study limitations are well-presented. The conclusions support the main study findings.
Overall, I find this manuscript interesting and easy to follow. However, for the international journal, the findings are too local, representing risky behaviours' features and differences across gender groups only in one country.
Author Response

(The authors gave the same response as above.)
